# StructCoder: Structure-Aware Transformer for Code Generation

## Abstract

There has been a recent surge of interest in automating software engineering tasks using deep learning. This paper addresses the problem of code generation where the goal is to generate target code given source code in a different language or a natural language description. Most of the state-of-the-art deep learning models for code generation use training strategies primarily designed for natural language. However, understanding and generating code requires a more rigorous comprehension of the code syntax and semantics. With this motivation, we develop an encoder-decoder Transformer model where both the encoder and decoder are explicitly trained to recognize the syntax and data flow in the source and target codes, respectively. We not only make the encoder structure-aware by leveraging the source code's syntax tree and data flow graph, but we also support the decoder in preserving the syntax and data flow of the target code by introducing two novel auxiliary tasks: AST (Abstract Syntax Tree) paths prediction and data flow prediction. To the best of our knowledge, this is the first work to introduce a structure-aware Transformer decoder that models both syntax and data flow to enhance the quality of generated code. The proposed StructCoder model achieves state-of-the-art performance on code translation and text-to-code generation tasks in the CodeXGLUE benchmark, and improves over baselines of similar size on the APPS code generation benchmark.

## 1 Introduction

Code generation is the problem of generating code given source code that is either imperfect or in a different language, or generating code from a natural language description. In this paper, we consider the problem of generating target code given source code in a different language (code translation) or a natural language description (text-to-code generation). Code translation has applications in migrating legacy codebases to contemporary programming languages and porting existing software to various other platforms (Rozière et al., 2020; Ahmad et al., 2021; Zhu et al., 2022). As developers often write code to solve a problem or implement logic that is stated in natural language, a good text-to-code generation model can help in speeding up the software development process (Ahmad et al., 2021). Traditional code translation tools have been designed using hand-crafted rules based on the Abstract Syntax Tree (AST) (Rozière et al., 2020). However, the design of such tools demands a lot of time and effort as it requires proficiency in both source and target languages (Zhu et al., 2022). Moreover, such tools are specific to the particular language pair they are designed for.

Since natural language generation using deep learning has achieved great success in recent years, it is natural to exploit similar deep learning based approaches for code generation as well. However, the code domain faces a unique set of challenges. Since the generated code is to be understood by a machine as opposed to a human, it is even more important for the generated code (compared to natural language) to adhere to a specific syntax. Moreover, since a minor change in code could alter its function, it is also critical to preserve the semantic information from the source code during translation. To generate syntactically correct code, some of the existing approaches for code generation leveraged the AST structure by learning to generate inorder traversal of AST (Li et al., 2018), learning to generate production rules for AST based on a grammar, encoding AST paths using RNNs (Alon et al., 2020), and using AST-based attention (Li et al.,

2018; Kim et al., 2021) in sequence models. Guo et al. (2021) hypothesize that Data Flow Graph (DFG), which contains more semantic information and is less complex than AST, is a more useful structure to learn code representations. They incorporate DFG into the Transformer encoder by appropriately masking the attention matrix. Our model, *StructCoder*, consists of a Transformer encoder that incorporates both syntax and data flow of source code by embedding root-leaf paths in the AST and using a modified self-attention framework, called *structure-aware self-attention.*

**Code generation heavily relies on the decoder to generate code that is syntactically correct while simultaneously preserving the semantics present in the input. Structcoder advances the state-of-the-art by incorporating a structure-aware Transformer decoder that is designed to preserve of syntax and semantics of the generated code.** None of the existing pretrained Transformer models constrain the generated code structure. In our work, we not only incorporate source AST and DFG into the encoder, but also enforce the decoder to learn the target syntax and data flow by introducing novel AST and DFG related tasks. Particularly, we train the decoder to predict all the root-leaf paths in the target AST and also to predict the DFG edges.

Similar to pretrained language models (Devlin et al., 2018; Liu et al., 2019; Radford et al., 2019; Yang et al., 2019), pretrained code models using Transformer (Feng et al., 2020; Ahmad et al., 2021; Lachaux et al., 2021; Zügner et al., 2021) have resulted in significant performance gains on code-related tasks. While some pretext tasks like Masked Language Modeling (MLM) and Replaced Token Detection (RTD) only pretrain the encoder, other pretext tasks like Denoising Autoencoding (DAE) and Back Translation (BT) jointly pretrain both the encoder and decoder. StructCoder falls in the latter category and is pretrained using a structure-based DAE task. Moreover, since the structure-based components introduced in this work can be added to any existing Transformer model, we may initialize most of the StructCoder weights using one of the pretrained code models to avoid pretraining from scratch which can be quite expensive. The main contributions of this work are listed below:

1. We develop a Transformer-based encoder-decoder model called StructCoder for code generation where both encoder and decoder are structure-aware. (a) The encoder incorporates AST's root-leaf path embeddings and a structure-aware self-attention framework to model source code structure. (b) The decoder is trained to recognize target syntax and data flow via two novel auxiliary tasks: AST paths prediction and data flow prediction.

2. We pretrain StructCoder using a structure-based DAE objective where the input code as well as its AST and DFG are partially corrupted and the model is trained to generate the original input code and also perform the auxiliary tasks.

3. Our experiments demonstrate that the proposed model achieves state-of-the-art performance on the code translation and text-to-code generation tasks in the CodeXGLUE (Lu et al., 2021) benchmark, and outperforms similarly sized baselines on the APPS code generation benchmark.

## 2 Related Work

### 2.1 Leveraging Structure to Generate Code

To leverage code structure in deep models, many approaches have utilized ASTs. Some approaches modeled code completion as a language modeling task by ordering the code tokens using a depth-first traversal of AST. Li et al. (2018) used an LSTM appended with parent-child attention while Alon et al. (2020) encoded each root-to-leaf path with an LSTM. Kim et al. (2021) used the Transformer to encode the sequenced AST by encoding AST paths into self-attention. For text-to-code generation, Rabinovich et al. (2017) proposed a modular decoder to recursively generate target AST. Yin & Neubig (2017); Brockschmidt et al. (2019); Sun et al. (2020) construct ASTs by generating production rules based on a grammar. Unlike these methods, we keep the conventional Transformer decoder architecture intact and introduce auxiliary structure-related components on top of the decoder's final hidden representations, so that StructCoder is trained to preserve target code structure while not requiring the generation of such structures (AST/DFG) during inference. Building on top of the conventional Transformer architectures not only allows us to utilize existing pretrained

models for better initialization but also makes the advances in the area of Transformers more easily applicable to our model.

## 2.2 Pretrained Transformers for Code

The recent state-of-the-art results on most natural language generation tasks are obtained by pretraining huge deep learning models on large datasets with carefully designed pretext tasks. Since code generation is very similar to text generation and there is abundant unsupervised code data available through open source code repositories, pretraining code generation models using similar pretext tasks has been successful. Most recent state-of-the-art pretrained models for code utilize the Transformer (Vaswani et al., 2017) architecture and are discussed below.

CodeBERT (Feng et al., 2020) performs encoder-only pretraining using Masked Language Modeling and Replaced Token Detection as pretext tasks on the CodeSearchNet dataset. Transcoder (Rozière et al., 2020) is an unsupervised translation model which pretrains both encoder and decoder using Denoising Autoencoding and Back-Translation with only monolingual datasets. PLBART (Ahmad et al., 2021) is pretrained with DAE objective using 680M Java and Python functions. DOBF (Lachaux et al., 2021) attempts to understand code structure with a deobfuscation pretext task where every occurrence of a sampled identifier is replaced by an uninformative token. Code Transformer (Zügner et al., 2021) modifies the attention computations in the encoder according to AST-based distances. CodeT5 (Wang et al., 2021) pretrains T5 model (Raffel et al., 2020) with code data in 8 programming languages. Different from PLBART which treats code data as plain sequences, CodeT5 includes identifier-aware objective in the training, which helps maintain the correctness of the code. However, CodeT5 does not include any structural information of the code in training. Zhu et al. (2022) improve code translation performance by introducing a fine-grained snippet-level translation task during pretraining. GraphCodeBERT (Guo et al., 2021) utilizes code structure in the form of Data Flow Graph (DFG) which contains semantic information as opposed to the syntatic information in AST. However, the decoder is completely unaware of the code structure in all of the above methods. *Our model advances the domain of code generation by being the first one to train a structure-aware Transformer encoder and decoder by modeling both syntax and data flow.* A summary of the pretext tasks and code structures used by the above Transformer-based methods along with our approach is provided in Table 1. To clarify the novelty of our work compared to other related methods like GraphCodeBERT, we provide a comparison of the two methods in the Appendix.

Table 1: A summary of the recent pre-trained models for code generation. (Abbreviations: DFG: Data Flow Graph, MLM: Masked Language Modeling, DAE: Denoising Autoencoding, RTD: Replaced Token Detection, BT: Back Translation, EP: DFG Edge Prediction, NA: Alignment prediction between code tokens and DFG nodes, DOBF: Deobfuscation, IT: Identifier Tagging, MSP: Masked Span Prediction, MIP: Masked Identifier Prediction, MuST: Multilingual Snippet Translation.)

| Model | Encoder-only pretraining | Encoder-Decoder pretraining | Encoder structure-awareness | Decoder structure-awareness |
|---|---|---|---|---|
| CodeBERT(Feng et al., 2020) | MLM, RTD | - | - | - |
| GraphCodeBERT(Guo et al., 2021) | MLM, EP, NA | - | DFG | - |
| Transcoder(Rozière et al., 2020) | MLM | DAE, BT | - | - |
| PLBART(Ahmad et al., 2021) | - | DAE | - | - |
| DOBF(Lachaux et al., 2021) | - | DOBF | - | - |
| CodeT5(Wang et al., 2021) | IT | MSP, MIP, NL-PL dual generation | Identifiers | Identifiers |
| MuST(Zhu et al., 2022) | - | DAE, MuST | - | - |
| StructCoder (ours) | | structure-based DAE, NL-PL dual generation | AST, DFG | AST, DFG |

# 3 StructCoder

StructCoder is a Transformer based encoder-decoder model where both encoder and decoder are structure-aware. We build our model using T5 architecture and add the relevant components for modeling code structure. For code inputs, the encoder (see Section 3.2) inputs the tokenized source code sequence along with its AST and DFG and employs structure-aware self-attention. The structure-aware decoder (see Section 3.3) simultaneously learns to generate the target code sequence as well as to perform target AST and DFG related tasks.

## 3.1 Preliminaries

A **Code** can be a function or a program, and is represented as a sequence of tokens $S = (s_1, ..., s_{|S|})$. A code $S$ has a corresponding **AST** represented as $\mathcal{T} = (N, N_{leaf}, r, p(.), L^{ast})$, where $N$ is the set of nodes in the AST, $N_{leaf} = \{l_1, ..., l_{|N_{leaf}|}\} \subset N$ is the subset of leaf nodes, $r \in N$ is the root node, $p : N - r \longrightarrow N$ is a mapping such that $p(n)$ denotes the parent of node $n$, and $L^{ast} \in \{0,1\}^{|S| \times |N_{leaf}|}$ is a linking matrix such that $L_{ij}^{ast} = 1$ if and only if token $s_i$ is part of leaf $l_j$. Each node $n \in N$ has a type denoted by $n.type$.

A code $S$ also has a corresponding **DFG** represented as $\mathcal{G} = (V, D, L^{dfg})$, where $V = \{v_1, v_2, ..., v_{|V|}\}$ is the set of variables extracted from code $S$, and $D \in \{0,1\}^{|V| \times |V|}$ is the adjacency matrix where $D_{ij} = 1$ if and only if value of $v_i$ is directly obtained from $v_j$, and $L^{dfg} \in \{0,1\}^{|S| \times |V|}$ is a linking matrix such that $L_{ij}^{dfg} = 1$ if and only if variable $v_j$ is derived from token $s_i$.

The goal of code translation is to transform a code $S = (s_1, ..., s_{|S|})$ in a source language to code $T = (t_1, ..., t_{|T|})$ in a different target language such that the translated code $T$ solves exactly the same problem as input code $S$ but in a different (target) language. In text-to-code generation, the goal is to generate target code $T$ from a natural language description.

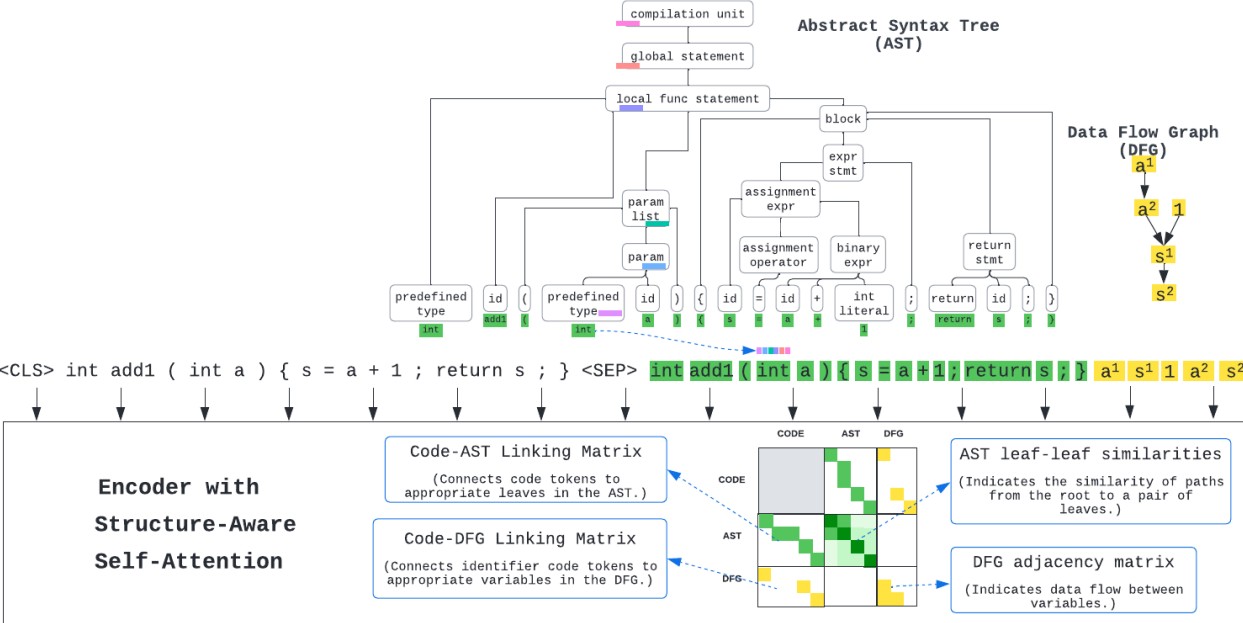

Figure 1: Structure-aware encoder: The input sequence to the encoder consists of source code concatenated with the AST leaves and DFG variables, where the AST leaves are embedded using the root-leaf paths in the AST. The modified structure-aware self-attention mechanism of this Transformer encoder utilizes code-AST/DFG linking information, leaf-leaf similarities in the AST, and the (asymmetric) DFG adjacency matrix to compute the attention matrix.

### 3.2 Structure-Aware Encoder

Given source code $S$, its corresponding AST $\mathcal{T}$, and DFG $\mathcal{G}$, the input sequence to the encoder is

$$\langle CLS \rangle, s_1, .., s_{|S|}, \langle SEP \rangle, l_1, ..., l_{|N_{leaf}|}, v_1, ..., v_{|V|}$$

which consists of the code tokens, special tokens $\langle CLS \rangle$ and $\langle SEP \rangle$, AST leaves, and DFG variables. For text input, the leaves and variables are simply ignored in the input. The encoder architecture is illustrated in Figure 1 and is described in detail below.

#### 3.2.1 Input Embedding

As StructCoder consists of a Transformer encoder, each token in the input sequence has to be embedded in $\mathbb{R}^d$. We embed the code tokens along with special tokens by using a lookup table, and use a default embedding for all DFG variables. The DFG information will be used by the encoder in structure-aware self-attention. We compute the embedding of a leaf $l$ in an AST as a function of the path from the root to the leaf $l$ in the AST.

Let $(r_1, r_2, ..., r_{|l|})$ be the nodes on the path from root $r = r_1$ to leaf $l = r_{|l|}$. We utilize node-type embedding $E_{type}(\cdot) \in \mathbb{R}^d$ to encode a node's semantics and a node-height embedding $E_{height}(\cdot) \in \mathbb{R}^d$ to encode the order of nodes on this path. The leaf embedding $E(l)$ is computed as

$$E(l) = \sum_{i=1}^{|l|} E_{type}(r_i.type) \odot E_{height}(|l| - i) \quad \in \mathbb{R}^d \tag{1}$$

where $\odot$ denotes element-wise multiplication.

#### 3.2.2 Structure-aware Self-attention

Since the input contains DFG and AST which consist of structural information, the traditional attention computation using (relative) positional embeddings which capture sequential ordering information is not sufficient. Hence, we propose structure-aware self-attention which computes attention scores between tokens based on the structural relations between them.

Code-code: Following T5, we compute attention scores (before softmax) between code tokens by adding the query-key dot product with weights $W_q, W_k \in \mathbb{R}^{d_k \times d}$ and a lookup embedding $\phi : \mathbb{Z}_{\geq 0} \longrightarrow \mathbb{R}$ for relative position. Denoting embedding of $x$ by $E_x$, we have

$$A(s_i, s_j) = E_{s_i}^T W_q^T W_k E_{s_j} + \phi(|i - j|) \tag{2}$$

Leaf-leaf: To calculate attention scores between leaves, we introduce a similarity-based transformation to replace the relative positional embedding in equation 2. Let $(r_1^i, ..., r_{|l_i|}^i)$ be the nodes on the path from root to leaf $l_i$. We define similarity between two leaves $l_i$ and $l_j$ as

$$sim(l_i, l_j) = log \left( \frac{\left( \sum_{k=1}^{min(|l_i|,|l_j|)} \mathbb{1}(r_k^i = r_k^j) \right)^2}{|l_i||l_j|} \right) \tag{3}$$

$$= 2\,log \left( \sum_{k=1}^{min(|l_i|,|l_j|)} \mathbb{1}(r_k^i = r_k^j) \right) - log\,|l_i| - log\,|l_j| \tag{4}$$

which is based on the number of common nodes on the paths from root to leaves $l_1$ and $l_2$. The *log* transformation is used to reduce the skewness of the distribution of similarity values. The attention scores between leaves are then computed as follows.

$$A(l_i, l_j) = E_{l_i}^T W_q^T W_k E_{l_j} + (w_a\,sim(l_i, l_j) + w_b) \tag{5}$$

where $w_a, w_b \in \mathbb{R}$.

Variable-variable: Following Guo et al. (2021), the attention scores between DFG nodes are computed using only the query-key dot product and are set to $-\infty$ if corresponding edges are absent in the DFG.

$$A(v_i, v_j) = \begin{cases} E_{v_i}^T W_q^T W_k E_{v_j} & \text{if } D_{ij} = 1 \\ -\infty & else \end{cases} \tag{6}$$

Code-leaf/variable: For interaction between code tokens and AST leaves (or DFG variables), we only compute the query-key dot product and do not use any positional information. Inspired by the work of Guo et al. (2021), we set the attention score to $-\infty$ for cases where the leaf (or variable) is not linked to the code token. We show the equations only for interactions between code tokens and leaves as those for interactions between code tokens and variables are similar.

$$A(s_i, l_j) = \begin{cases} E_{s_i}^T W_q^T W_k E_{l_j} & \text{if } L_{ij}^{ast} = 1 \\ -\infty & else \end{cases} ; \quad A(l_j, s_i) = \begin{cases} E_{l_j}^T W_q^T W_k E_{s_i} & \text{if } L_{ij}^{ast} = 1 \\ -\infty & else \end{cases} \tag{7}$$

The special tokens $\langle CLS \rangle$ and $\langle SEP \rangle$ are treated just like code tokens and are assumed to be linked to all leaves and variables.

### 3.3 Structure-Aware Decoder

The decoder in StructCoder constitutes the original T5 decoder with additional layers at the end for AST paths prediction and data flow prediction tasks that are introduced in this section. Figure 2 illustrates the structure-aware decoder which predicts the next target code token along with the AST root-leaf path to this token and the data flow relations between this token and all past tokens. The addition of these auxiliary tasks does not increase the number of generated tokens, which is important since the decoding is done in an autoregressive manner.

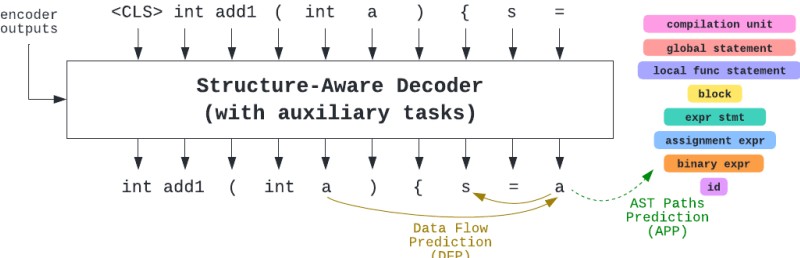

Figure 2: Structure-aware decoder generates the next token in the target code as well as predicts the node types on the root-leaf path to the leaf containing this token in the target AST and also the DFG edges incident on this token.

Let $h_1, h_2, ..., h_{|T|}$ be the hidden states generated by the Transformer decoder. Decoders of existing transformer models including T5 employ a linear layer with weights $W \in \mathbb{R}^{|\mathcal{V}| \times d}$ followed by softmax transformation to extract a probability distribution $p_i$ on the token vocabulary space $\mathcal{V}$ for the $i^{th}$ position.

$$p_i = softmax\,(W h_i) \tag{8}$$

And the sequence generation task is trained using language modeling loss as shown below for one sample.

$$\mathcal{L}_{lm} = -\sum_{i=1}^{|T|} log\ p_i(t_i) \tag{9}$$

where $p_i(t_i)$ refers to the predicted probability for true target token $t_i$ at the $i^{th}$ position.

In addition to sequence generation, StructCoder also learns target syntax using AST paths prediction task, and learns to match target DFG using a data flow prediction task.

### 3.3.1 AST Paths Prediction (APP)

In this task, the goal is to enable the decoder to be aware of all root-leaf paths in the target AST. Since the type attribute of a node captures important syntactic information, we predict the type of each ancestor on each root-leaf path.

Let $l_i$ be the leaf node containing the $i^{th}$ target token $t_i$ and let $(r_1^i, ..., r_{|l_i|}^i)$ be the nodes on the root-$l_i$ path. To predict type of node $r_k^i$ (which is at height $|l_i| - k$ in the tree), we use a linear layer with weights $W_{ast\,(|l_i|-k)} \in \mathbb{R}^{|\mathcal{Y}| \times d}$ followed by a softmax transformation to predict a probability distribution on the set of node types $\mathcal{Y}$.

$$p_{ik}^{ast} = softmax(W_{ast\,(|l_i|-k)}h_i) \tag{10}$$

The APP loss for a sample is given by

$$\mathcal{L}_{app} = -\sum_{i=1}^{|T|}\sum_{k=1}^{|l_i|} log\, p_{ik}^{ast}(r_k^i.type) \tag{11}$$

### 3.3.2 Data Flow Prediction (DFP)

In this task, the decoder learns to predict all the data flow edges in target code. The probability $p_{ij}^{dfg}$ of data flow from $j^{th}$ to $i^{th}$ position in target code sequence is computed using an asymmetric transformation (since data flow is directed) as

$$p_{ij}^{dfg} = \sigma\left(h_i^T U_{dfg}^T V_{dfg} h_j + u_{dfg}^T h_i + v_{dfg}^T h_j + w_{dfg}\right) \tag{12}$$

where $\sigma(.)$ denotes the sigmoid function. Suppose $\mathcal{G} = (V, D, L)$ is the true target DFG. There is a data flow from $j^{th}$ to $i^{th}$ position in target sequence if and only if "target DFG contains variables $v_{j'}$, $v_{i'}$ such that variable $v_{j'}$ is derived from $t_j$, variable $v_{i'}$ is derived from $t_i$, and value of variable $v_{i'}$ is derived from $v_{j'}$". Thus, the DFP loss for a sample can be written as

$$\mathcal{L}_{dfp} = -\sum_{i=1}^{|T|}\sum_{j=1}^{|T|} \left\{ \mathbb{1}(cond)\, log\, p_{ij}^{dfg} + \mathbb{1}(\neg\, cond)\, log\, (1 - p_{ij}^{dfg}) \right\}$$
$$\text{where} \quad cond = (\exists\, v_{i'},\, v_{j'} \in V \text{ such that } D_{i'j'} = L_{ii'}^{dfg} = L_{jj'}^{dfg} = 1) \tag{13}$$

The overall loss function for training StructCoder (given below) is a combination of the language modeling objective, and the APP and DFP losses with weights $\alpha_1$ and $\alpha_2$ that may be fixed or trainable.

$$\mathcal{L} = (3 - \alpha_1 - \alpha_2)\mathcal{L}_{lm} + \alpha_1\mathcal{L}_{app} + \alpha_2\mathcal{L}_{dfp} \tag{14}$$

### 3.4 Pretraining

We pretrain StructCoder on a subset of CodeSearchNet[1] (Husain et al., 2019) dataset with structure-based DAE task along with NL-PL bimodal dual generation to generate code from text and vice-versa. For the denoising task, we corrupt random spans in the code sequence as well as drop some DFG variables and AST leaves in the input to encoder, and the model is trained to predict the uncorrupted code along with the node types on AST root-leaf paths and data flow edges. We initialize our model for pertaining with CodeT5's weights (for faster pretraining) except for the AST and DFG related weights which are randomly initialized. More details on our pretraining method are provided in the Appendix.

---

[1] https://github.com/github/CodeSearchNet

## 4 Experiments

We evaluate StructCoder on the code translation and text-to-code generation tasks from the CodeXGLUE [2] (Lu et al., 2021) benchmark, and on the text-to-code generation task from the APPS benchmark (Hendrycks et al., 2021), and compare with previously published results on these tasks. For CodeXGLUE tasks, we use the metrics from the CodeXGLUE leaderboard which include (i) BLEU (Papineni et al., 2002) score which measures n-gram overlap, (ii) exact match (xMatch) which checks if the prediction is the same as ground truth, and (iii) CodeBLEU (Ren et al., 2020) which combines BLEU score with keywords-based weighted n-gram match as well as syntax and semantic matches based on AST and DFG. APPS evaluates generated codes based on test cases where the evaluation metrics include (i) 'test case average' which is the average percentage of test cases passed, and (ii) 'strict accuracy' which is the percentage of generated codes that pass all test cases. The implementation details are provided in the Appendix and the code is available in the supplementary materials.

### 4.1 Code Translation

The code translation dataset from CodeXGLUE consists of two tasks for translating between Java and C# functions in either direction and contains 10K training samples, 500 validation samples, and 1000 test samples. Table 2 presents the results of StructCoder alongside the baselines on the two code translation tasks. The Naive Copy baseline simply copies source code to target and the Transformer model involves no pretraining. RoBERTa (code) (Lu et al., 2021), CodeBERT, and GraphCodeBERT involve encoder-only pretraining while PLBART and CodeT5 incorporate encoder-decoder pretraining like StructCoder. StructCoder achieves the best results on the two translation tasks which can be attributed to the structure-aware encoder-decoder design of our model. From Table 2, we observe that the encoder-decoder pretraining of PLBART, CodeT5, and StructCoder is very beneficial to code translation. Also, the encoder-only pretrained models improve over Transformer by a huge margin. GraphCodeBERT which utilizes DFG offers minor improvements over CodeBERT and we also observed in our ablation study that DFG-related components contribute less to the performance gains of StructCoder compared to AST-related components.

Table 2: Results on code translation tasks from CodeXGLUE benchmark. (*Since CodeT5 is a competitive baseline and did not report CodeBLEU in their paper, we tested this model using their finetuned checkpoint and provided the results.)

|  | Java-C# | | | C#-Java | | |
|---|---|---|---|---|---|---|
|  | BLEU | xMatch | CodeBLEU | BLEU | xMatch | CodeBLEU |
| Naive Copy | 18.54 | 0.00 | 42.20 | 18.69 | 0.00 | 34.94 |
| Transformer | 55.84 | 33.00 | 63.74 | 50.47 | 37.90 | 61.59 |
| RoBERTa (code) | 77.46 | 56.10 | 83.07 | 71.99 | 57.90 | 80.18 |
| CodeBERT | 79.92 | 59.00 | 85.10 | 72.14 | 58.80 | 79.41 |
| GraphCodeBERT | 80.58 | 59.40 | - | 72.64 | 58.80 | - |
| PLBART | 83.02 | 64.60 | 87.92 | 78.35 | 65.00 | 85.27 |
| CodeT5* | 83.88 | 64.70 | 87.38 | 79.71 | 67.50 | 85.51 |
| StructCoder | **85.03** | **66.60** | **88.41** | **80.73** | **67.70** | **86.10** |

### 4.2 Text-to-Code Generation

The text-to-code generation task from CodeXGLUE uses the CONCODE (Iyer et al., 2018) dataset and the goal here is to generate a Java function given a natural language description. This dataset contains 100K training samples, 2K validation samples, and 2K test samples. Table 3 presents the results of our model alongside the baselines on the text-to-code generation task. Among the baselines, GPT-2 (Radford et al., 2019) is pretrained on natural language to predict next token, CodeGPT (Lu et al., 2021) is pretrained from scratch like GPT-2 but using code data, CodeGPT-adapted (Lu et al., 2021) is pretrained from GPT-2

---

[2]https://github.com/microsoft/CodeXGLUE

initialization using code data, and CoTexT (Phan et al., 2021) pretrains the T5 model further on code data using MSP objective. The decoder-only baselines which include GPT-2 based models are outperformed by the rest which are all encoder-decoder models. StructCoder again achieves the best performance on all metrics for the text-to-code generation task.

Table 3: Results on text-to-code generation task from CodeXGLUE benchmark.

|                | BLEU | xMatch | CodeBLEU |
|----------------|-------|--------|----------|
| GPT-2          | 25.37 | 17.35  | 29.69    |
| CodeGPT        | 28.69 | 18.25  | 32.71    |
| CodeGPT-adapted| 32.79 | 20.10  | 35.98    |
| PLBART         | 36.69 | 18.75  | 38.52    |
| CoTexT         | 37.40 | 20.10  | 40.14    |
| CodeT5         | 40.73 | 22.30  | 43.20    |
| StructCoder    | **40.91** | **22.35** | **44.77** |

Table 4: Results on the APPS dataset along with model size in #billion parameters. The results for GPT-2 models were obtained from Hendrycks et al. (2021).

|             |            | Test case average | | | Strict accuracy | | |
|-------------|------------|-------|-----------|-------------|-------|-----------|-------------|
|             | Model size | Intro | Interview | Competition | Intro | Interview | Competition |
| GPT-2       | 0.1B       | 5.64  | 6.93      | 4.37        | 1     | 0.33      | 0           |
| CodeT5      | 0.2B       | 5.50  | 5.06      | 2.33        | 0.6   | 0.67      | 0           |
| StructCoder | 0.2B       | **10.01** | 7.09  | 3.57        | **1.8** | **0.73** | **0.2**    |
| GPT-2       | 1.5B       | 7.40  | **9.11**  | **5.05**    | 1.3   | 0.70      | 0           |

APPS (Hendrycks et al., 2021) is a text-to-code generation benchmark in python which evaluates generated codes based on test cases. The inputs here contain detailed questions and possibly some starter code as well. The dataset contains 10K problems equally divided into train and test splits. The test set contains 1K introductory level, 3K interview level, and 1k competition level problems. Table 4 shows the results of StructCoder, CodeT5, and GPT-2 (Hendrycks et al., 2021) models of two sizes. These GPT-2 models were pretrained exclusively on python code from GitHub which gives them an edge in this particular task. The 'strict accuracy' metric is more important than the 'test case average' as it does not give partial credit to a generated code that does not pass all test cases. StructCoder achieves the best 'strict accuracy' on all subsets, notably outperforming the bigger GPT-2 model which is about 7 times the size of StructCoder.

### 4.3 Model Analysis

#### 4.3.1 Ablation Study

To emphasize the importance of the novel structure-based components introduced in this work, we conducted an ablation study on the two code translation tasks from CodeXGLUE. For this experiment, we used a smaller T5 architecture with hidden dimension 256, 5 encoder and decoder layers, and 8 heads in each multi-head attention layer. The models we tested here include the smaller T5 model (i) without any structure-awareness which is our baseline, (ii) with enabling AST or DFG related components in encoder or decoder, (iii) with enabling all the structure-based components, and (iv) adding structure-based DAE pretraining to the previous one. Note that the first three cases do not involve any pretraining. We report the xMatch, and CodeBLEU and its different components. The results are shown in Table 5.

On both tasks, including AST and DFG in both encoder and decoder yields the best results on all metrics. When considering the effect of individual components, we see that adding AST to encoder works the best for Java-C# translation while adding AST to decoder fares well for C#-Java translation. Each component individually improves the translation performance significantly over the baseline except for the case of adding DFG to encoder for C#-Java translation. In general, adding DFG or AST components improves not just Data Flow match or AST match but all the metrics. The model with structure-based DAE pretrained in

Table 5 has been pretrained from scratch on just 30K samples for 15K steps (where each step is a gradient update using a batch of 32 samples). While all other models in this study were trained for 50K steps, this model has been finetuned for just 10K steps, keeping the architecture and other training parameters the same across all models. We observed that structure-based DAE pretraining led to significant performance gains on both tasks.

Table 5: Results on Java-C# and C#-Java translation by adding the proposed structure-based components to a smaller T5 model. The **best** results are in bold and the second best are underlined. ('i/p' and 'o/p' indicate whether the structure was included in the encoder and decoder, respectively.)

| Enabled | xMatch | BLEU | Weighted BLEU | AST match | Data Flow match | CodeBLEU |
|---|---|---|---|---|---|---|
| Java-C# translation | | | | | | |
| No structure (baseline) | 43.90 | 62.30 | 63.60 | 78.82 | 73.79 | 69.62 |
| DFG (i/p) | 47.20 | 65.59 | 66.72 | 80.04 | 75.66 | 72.00 |
| DFG (o/p) | 48.10 | 64.87 | 66.12 | 79.88 | 75.26 | 71.53 |
| AST (i/p) | 51.10 | 69.92 | 70.93 | 82.89 | 77.97 | 75.42 |
| AST (o/p) | 46.00 | 64.16 | 65.34 | 80.02 | 75.45 | 71.24 |
| DFG (i/p,o/p), AST (i/p,o/p) | 51.20 | 70.86 | 71.82 | 83.87 | 79.41 | 76.49 |
| DFG(i/p,o/p), AST(i/p,o/p), & structure-based DAE pt | **53.80** | **76.86** | **78.07** | **87.07** | **85.00** | **81.75** |
| C#-Java translation | | | | | | |
| No structure (baseline) | 40.20 | 53.20 | 54.56 | 75.40 | 64.20 | 61.84 |
| DFG(i/p) | 27.10 | 41.64 | 43.20 | 70.19 | 58.63 | 53.41 |
| DFG(o/p) | 43.10 | 56.64 | 57.90 | 77.24 | 66.52 | 64.57 |
| AST(i/p) | 45.90 | 59.25 | 60.30 | 79.12 | 68.31 | 66.74 |
| AST(o/p) | 49.50 | 63.70 | 64.79 | 81.84 | 72.89 | 70.80 |
| DFG(i/p, o/p), AST(i/p, o/p) | 51.20 | 66.12 | 66.99 | 83.79 | 74.30 | 72.80 |
| DFG(i/p,o/p), AST(i/p,o/p), & structure-based DAE pt | **55.10** | **73.53** | **74.41** | **87.30** | **83.80** | **79.76** |

### 4.3.2 Auxiliary Tasks

We measure the performance of StructCoder on the auxiliary tasks of APP (AST Paths Prediction) and DFP (Data Flow Prediction) as follows. When predicting the next target token, we use the ground truth for target sequence until the previous step as input to the decoder. The decoder then predicts the next token as well as the DFG edges incident on this token and the types of nodes on the path from root to the leaf node containing this token in the AST. On Java-C# translation, StructCoder achieves 76.6% accuracy on APP task and 74.8% average precision on DFP task where positive class prevalence is just 0.8%. On C#-Java translation, StructCoder achieves 76.8% accuracy on APP task and 33.6% average precision on DFP task where positive class prevalence is just 0.5%. For both the translation tasks, there are 291 classes for node type in APP task.

### 4.3.3 Case Study

Figure 3 shows an example from Java-C# translation task with predictions from StructCoder and the best baseline CodeT5. We observe that our structure-aware encoder-decoder architecture is able to correctly generate the target code where CodeT5 fails, which can be explained by inspecting the predictions of the two auxiliary tasks. Referring to Figure 3, CodeT5 generates both the 'for' loops with index 'i', leaving variable 'c' undefined. It also misses the first 'if' statement and creates a syntax error from unbalanced

Figure 3: Case study: An example from Java-C# translation task where StructCoder is able to accurately predict the target code while CodeT5 fails. (Red text indicates errors made by CodeT5 and blue text indicates correctly predicted code by StructCoder where baseline generates errors. The blue arrows show some of the correctly predicted (probability $\geq 97^{th}$ percentile) data flow edges relevant to the colored text.)

braces. On the other hand, StructCoder correctly generates the for loops by defining variable 'c' and the model predicts (with probability greater than or equal to $97^{th}$ percentile) most of the DFG edges incident on the variable 'c' inside these for loops and also in the first 'if' statement. Also, for token '[]' in args, the correct parent node type 'array rank specifier' is in the top two predicted node types. More examples are included in the Appendix.

### 4.3.4 Complexity Analysis

For the translation experiments, the input length to the encoder approximately doubles by adding the structure-related tokens. Since the time (for sequential computation) and memory complexity of self-attention is quadratic in input length, the time and memory requirements of the self-attention module increase by about four-fold upon adding structure in our translation experiments. However, since self-attention is parallelizable (Vaswani et al., 2017), we do not observe a significant increase in inference time. We provide more details on inference times and discuss model sizes in the Appendix.

## 5 Conclusion

This work proposes a structure-aware Transformer encoder-decoder model called StructCoder for code generation. Our encoder modifies traditional input embeddings and employs a structure-aware self attention mechanism to model AST and DFG relations in source code, and the decoder is trained to recognize target syntax and data flow using two novel auxiliary tasks to predict the node types on all root-leaf AST paths and data flow edges of target code. We also pretrained our model using a structure-based DAE task to improve its performance. Experiments on code translation and text-to-code generation tasks demonstrate the performance gains of StructCoder over state-of-the-art baselines. We believe that this work would encourage future research in this field to give careful consideration to code structure while building models for code generation.

## 6 Broader Impact

Although automated code generation can potentially benefit the development and migration of software, there are risks associated with it. First of all, the model is not capable of taking into consideration constraints like security, efficiency, and modularization when generating code. Thus, deploying model-generated code can introduce vulnerabilities in complex systems, and increase the energy cost and emissions. Furthermore, the maintenance of the generated code can be challenging if it is less modularized.

Second, the performance improvements of the code generation models largely rely on scaling-up of both the model and the training, which require significant amount of computational resources. Individuals, small organizations, and academic institutes usually cannot afford the large-scale training of such models, while big companies have a natural advantage in this aspect. Therefore, the advances in this domain might benefit the big businesses more than general audience, which limits the societal value of it.

Third, the deployment of automated code generation tools requires reforming the current skill sets for software engineering jobs. But once the users are well-educated about the usage and maintenance of such systems and the security risks associated with them, the software development process should become more efficient.

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

## A StructCoder vs GraphCodeBERT

It should be noted that there are several significant differences between Structcoder and other prominent methods in the literature like GraphCodeBERT. The only similarity between StructCoder and GraphCodeBERT with respect to modeling structure is the approach for encoding source DFG in the encoder. The differences are listed below:

- While the pretext tasks of GraphCodeBERT include (a) alignment prediction between source code tokens and source DFG nodes and (b) edge prediction between source DFG nodes, StructCoder includes a (target) Data flow prediction task which directly predicts the data flow relations between (target) code tokens generated by the decoder.

- Our approach to modeling AST in the encoder differs from that of DFG as follows: (a) Instead of adding a separate input token per AST node, we only add an input token per AST leaf, which contains the input sequence length to an extent. (b) While tokens of DFG nodes use a default node embedding, the AST leaves are embedded using the root-leaf paths in the tree. (c) While node-node attention between DFG nodes is based on DFG edges, the leaf-leaf attention between AST leaves is based on their positional similarity/proximity in the tree.

- GraphCodeBERT is an encoder-only model which requires a non-pretrained decoder for code generation. In contrast, StructCoder jointly pretrains the encoder and decoder using a novel structure-based DAE objective.

# B   Model Analysis

## B.1   Model size

Table A1 shows the number of parameters in different pretrained models for code. Note that StructCoder is built by adding additional components to CodeT5 for modeling AST and DFG in input and output, with majority of additional parameters coming from the encoder's AST leaves embedding module (381K) and the classification layer of APP (AST Paths Prediction) task (743K).

Table A1: Number of parameters in various pretrained models

| Pretrained model | # parameters |
| --- | --- |
| CodeBERT | 125M |
| GraphCodeBERT | 125M |
| CodeGPT-small-java | 126M |
| PLBART | 139M |
| CodeT5 | 223M |
| CoTexT | 223M |
| StructCoder | 224M |

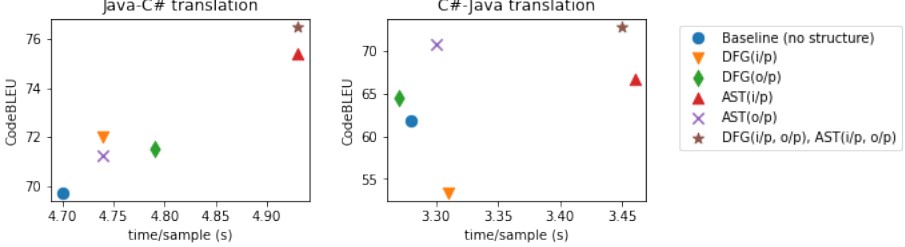

Figure A1: Scatter plots of CodeBLEU vs inference time for models used in the ablation study in the main paper. For measuring inference time, we generated predictions for 50 samples using a batch size of 1 on the CPU since using a 48GB GPU did not give any noticeable differences among models' inference times.

## B.2   Inference time

To analyze the impact of adding the proposed structure-based components on the overall computational complexity experimentally, we measured the inference times for the models used in the ablation study for the two translation tasks. We report the results by running inference on the CPU for 50 samples using a batch size of 1. For the 50 examples used here, the average input sequence length is 50.06 (59.48) and 100.10 (118.50) for the baseline and fully structure-aware model respectively, for Java-C# (C#-Java) translation. Figure A1 shows scatter plots of CodeBLEU vs average inference time per sample. Note that ignoring size and pretraining, "baseline" is equivalent to CodeT5 and enabling "DFG(i/p, o/p), AST(i/p, o/p)" is equivalent to StructCoder. During inference, since we do not perform auxiliary tasks, enabling AST or DFG structures in decoder should not impact inference time. Figure A1 also shows that adding DFG to encoder leads to negligible difference while adding AST to encoder increases inference time per sample by approximately only 5% on the two translation tasks though the average input length to the encoder increases by 80-85%. To support this finding, we ran beam search decoding on the original T5 model for similar input lengths where we found a similar trend. The increase in inference time being much less than expected may be due to implementation-specific details of pytorch/huggingface which are out of scope for this work.

## C   Implementation Details

We use the CodeT5 tokenizer with a vocabulary size of 32,100. As we build upon CodeT5 architecture, both the encoder and decoder of StructCoder contain 12 T5 blocks with hidden dimension 768, and 12 attention heads in each block. During implementation, we only used first 16 bits of the last hidden representation from the decoder to predict DFG links and the next 128 bits for AST paths prediction. This is done because the model learns DFP task more easily than APP task and using few bits for these auxiliary tasks prevents overfitting on these tasks. If the weights of auxiliary task losses in the overall loss function are set to be trainable, we initialize parameters $a_1, a_2 \in \mathbb{R}$, and compute $\alpha_1 = \sigma(a_1)$, $\alpha_2 = \sigma(a_2)$ to constrain the loss weights $\alpha_1, \alpha_2$ to lie in $[0, 1]$. All experiments used AdamW optimizer.

**Pretraining**

We pretrain StructCoder on a structure-based DAE task along with NL-PL bimodal dual generation to generate code from text and vice-versa. For the denoising task, we first corrupt random spans in a code sequence by replacing with $\langle MASK \rangle$ or a random token, or deleting it. The span lengths are sampled from a Poisson distribution of mean 3.5 and we corrupt 35% of the code tokens in total, similar to Ahmad et al. (2021). To improve the understanding of code structure, we also randomly drop 35% of the DFG variables and AST leaves, and 35% of the ancestors for each leaf from the input to StructCoder. The model is then trained to predict the uncorrupted code along with the AST root-leaf paths and data flow edges. We pretrain StructCoder on a randomly chosen subset of 300K samples from CodeSearchNet[3] (Husain et al., 2019) belonging to four popular languages: Python, PHP, JavaScript, and Java. We initialized the non-structure-based weights with CodeT5 pretrained model for faster learning, and ran our pretraining with a batch size of 32 for 12K steps, with a learning rate of 5e-5.

**Finetuning**

For code translation, we ran finetuning with a batch size of 25 for 22K steps. For text-to-code generation using CONCODE dataset, we ran finetuning with a batch size of 32 for 100K steps. To finetune on APPS dataset, we used a batch size of 16 for 15K steps. The learning rate set of 5e-5 for tasks in CodeGLUE and 5e-4 for the APPS benchmark. We set $a_1, a_2$ to be trainable and clipped them to have a max value of -4 during finetuning, except for APPS where we fixed $\alpha_1 = \alpha_2 = -2$. For new AST node types seen during finetuning, we initialized the weights corresponding to these new node types randomly. We used beam search with a beam size of 10 for decoding in all finetuning tasks except for the APPS dataset where the beam size was set to 5. We ran validation every 500 steps and chose the checkpoint with the best BLEU score on the validation set for testing. For APPS, which has no validation set, the checkpoint at the end of the training was used for inference. Since CodeT5 does not have published results on the APPS dataset, we finetuned it using the same hyperparameters used by our model.

For the ablation study, we reported most of the hyperparameters used in the main text. Among the remaining ones, the learning rate was set to 0.001 except for the pretraining stage which used a learning rate of 5e-4, and the beam size was set to 5. The loss weights of auxiliary tasks were fixed to $a_1 = a_2 = -1$ for C#-Java translation where both auxiliary tasks are used, and we fix $a_1 = a_2 = -2$ in all other cases. For C#-Java translation, we experimented with values of $a_1 = a_2 = -1$ and $a_1 = a_2 = -2$.

**Sequence lengths**

To facilitate minibatch training with available resources, we set max no. of DFG variables in input to 75, max no. of AST leaves to 250, and max root-leaf path length to 12 (by trimming paths from the root's side). We set the max source length (no. of code tokens when input is code) to 400 for pretraining, 320 for translation, 325 and 1024 for text-to-code generation on CONCODE and APPS. We set max target length to 400 for pretraining, 320 for Java-C# translation, 256 for C#-Java translation, 155 and 550 for text-to-code generation on CONCODE and APPS respectively. We used the same sequence lengths as StructCoder

---

[3]https://github.com/github/CodeSearchNet

for finetuning CodeT5 on APPS. The results of CodeT5 on CodeXGLUE tasks were borrowed from Wang et al. (2021) where the max source and target lengths were set to 512 and 256, respectively. On the code translation tasks, GraphCodeBERT (Guo et al., 2021) sets max source and target lengths to 256 and max no. of DFG variables to 64.

**Other details**

All the hyperparameters discussed above were set either based on CodeT5's implementation, or in rare cases, by observing the progression in validation performance for a few steps, or by choosing the ones with best validation performance after a few trials. We used Pytorch (Paszke et al., 2019) and Huggingface[4] (Wolf et al., 2020) libraries to implement our model. The code for generating ASTs and DFGs is built using tree-sitter [5] and is also adapted from `https://github.com/microsoft/CodeBERT/tree/master/GraphCodeBERT`. The random generators were seeded for each experiment in the 'set_seed()' function. We ran our experiments on an Ubuntu 18.04 server with 4 RTX 8000 GPUs with 48GB memory on each GPU. The code is included in the supplementary material and will be made public after publishing this work.

## D Examples

In this section, we illustrate a few examples of text-to-code generation along with the predicted DFG links and AST paths (see Figures A2-A4). The DFG predictions are visualized as a matrix where the $ij^{th}$ cell denotes the probability of data flow from $j^{th}$ to $i^{th}$ token. To visualize predicted AST paths, for each predicted token, we indicate the predicted node types on the path starting from the root (top) to the leaf

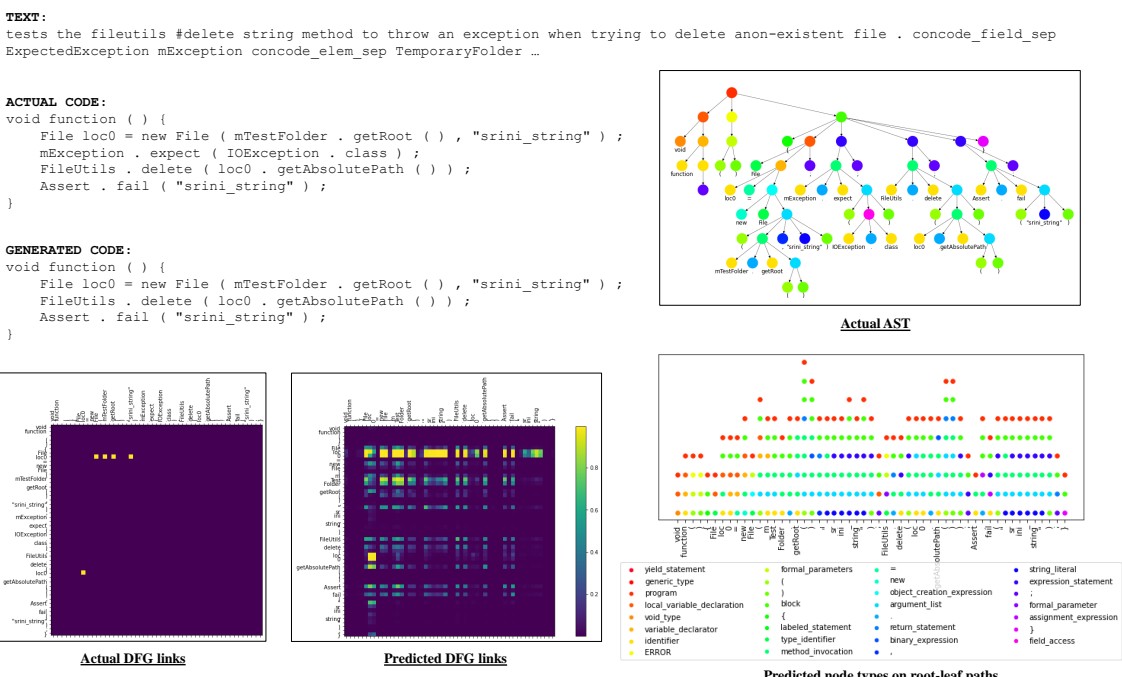

Figure A2: An example from the concode dataset with BLEU=78.85.

---

[4]https://huggingface.co/transformers/

[5]https://github.com/tree-sitter/py-tree-sitter

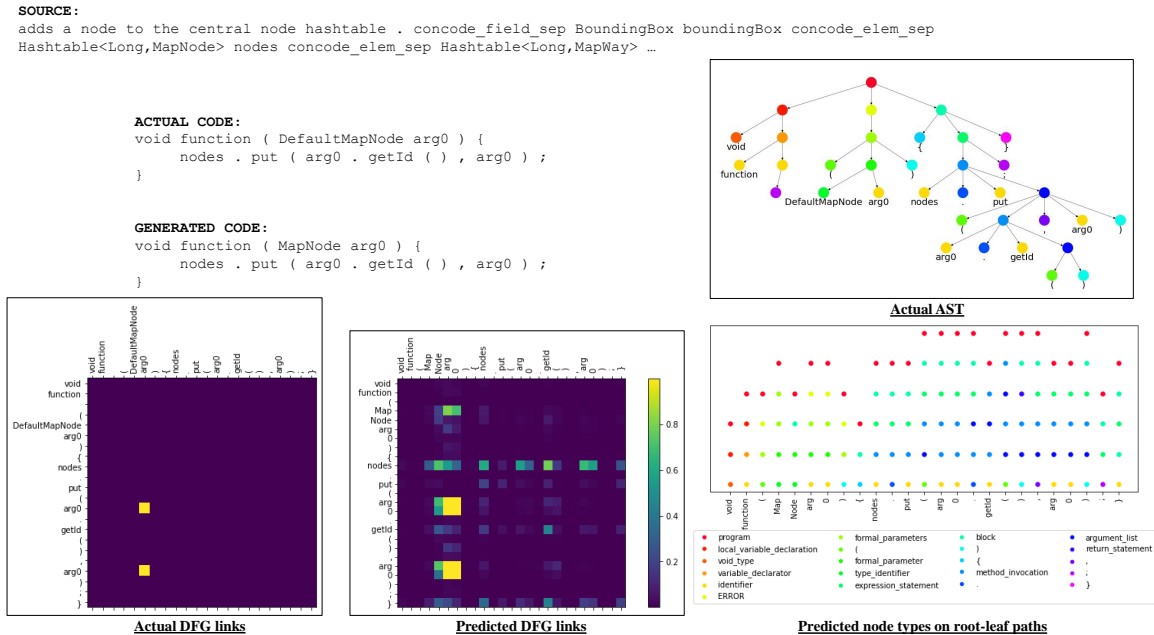

Figure A4: An example from the concode dataset with BLEU=87.25.

