# OpenReview forum: "StructCoder: Structure-Aware Transformer for Code Gener-ation"
_TMLR — Rejected by TMLR_

### Review · Reviewer_b4TR · 2022-11-14

**Summary Of Contributions:**

The paper addresses the tasks of code translation and code generation from natural language.
While most state-of-the-art approaches treat code as flat text (in both the encoder and the decoder), this paper proposes to encode both the AST and data flow information, and also predict them in the decoder.
Concretely, the standard token input is appended with (1) The leaves of the AST, where each of them is encoded as a root-to-leaf path; and (2) data flow nodes.
The attention mechanism is modified: the attention between AST leaves is based on their proximity in the AST, and the attention between dataflow nodes is "GNN-style" - where only neighboring variables attend to each other.
The model is initialized from the CodeT5 weights, and adds a few additional weights.
The resulting StructCoder model improves over the strong CodeT5 baseline in code translation and text-to-code tasks from CodeXGLUE.

**Audience:**

Yes

**Broader Impact Concerns:**

-

**Claims And Evidence:**

No

**Requested Changes:**

## Major changes
* Costs: Please detail the training and inference costs compared to the base CodeT5, in terms of memory, time, batch size, and throughput.
* Please elaborate on whether the evaluation is fair. Specifically, it is crucial to pretrain the base CodeT5 for (at least) the same time and on the same training data as the proposed StructCoder.
* Scalability - the model (0.2B) was shown to be competitive with a GPT-2 1.5B. Can it be applied to the GPT-2 1.5B and perform even better?
* Additional ablation study of "only-AST" and "only-DFG" (in both encoder and decoder).
* Additional baselines, if possible.

## Presentation Comments and questions:
* The first paragraph in the Introduction contains many claims that are understandable, but still require citations to justify.
* The "AST Path Prediction" is also referred to as "APP", which is confusing because the paper also mentions the APP dataset.
* Parsing - how did the authors deal with examples that do not syntactically parse in the Java-C# dataset?
* AST Path Prediction - are all nodes in a single path predicted at the same time, independently of each other?
* Figure 1 is unreadable, some text is too tiny

**Strengths And Weaknesses:**

### Strengths
* The motivation is clear, and the idea of encoding syntax and data flow is appealing.
* The results improve over the baselines in two tasks.
* In the APPS dataset, the proposed StructCoder is competitive with a larger GPT-2 1.5B model.

### Weaknesses
* **The computational cost is unclear** - there are several sources of additional cost that the authors did not mention:
    * **The >2x longer input, which increases compute quadratically** - the authors only mentioned that they "did not observe a significant increase in inference time" - but even at inference time, this must have decreased the possible batch size by ~4x! How did this effect training time? How smaller did the authors have to decrease the batch size, and how slower was the training?
    * **The encoding of the AST inputs** - What is the overhead of encoding the root-to-leaf paths?
    * **Data Flow Prediction** - what is the overhead of predicting edges? What is the number of possible edges?
    * **AST Path Prediction** - What is the memory and time overhead of predicting all nodes in all paths?
* **Soundness of Evaluation** - I am not sure about the soundness of the evaluation. Concretely, I am not sure that all models and baselines were given the same chances.
    * The proposed StructCoder was pretrained on more data and for more time than the standard CodeT5.
    * The proposed StructCoder was pretrained on Denoising Auto Encoding (DAE) task, which can be applied to the base model as well.
* **Ablation Study** - The ablation study is thorough, but misses the following ablation: what if the model was pretrained+trained with only-AST in the encoder and decoder? what if the model was pretrained+trained with only-DFG in the encoder and decoder?
* **Additional Possible baselines**:
    * Codex and its smaller versions through the OpenAI API, even though it is larger, just to understand the upper bound.
    * CodeGen and its smaller versions ( https://arxiv.org/pdf/2203.13474.pdf )
    * PolyCoder ( https://arxiv.org/pdf/2202.13169.pdf )

---

> ### Author Response · Authors · 2022-12-28
> **Author's response**
>
> - Fairness of evaluation : StructCoder was pretrained on a small subset (just 3.6%) of the data used for CodeT5 pretraining. And we did the extra pretraining for StructCoder to pretrain the weights of newly introduced structure-related components in StructCoder. CodeT5 was pretrained on 8.35M samples for 150 epochs using a batch size of 1024. Compared to this, our extra pretraining on 300K samples using a batch size of 32 for 12K steps is negligible. Thus, we think that the comparison with CodeT5 is fair.
>
> - Thank you for the questions about costs and suggestions for additional baselines and ablations. If given more time, we can do a more thorough analysis of costs and run the suggested experiments. (The results in the appendix only measured inference time using the same batch size for all models.)
>
> - Scalability: With the available resources, it is difficult for us to train a 1.5B parameter GPT model. Though we think that adding the proposed structure related components to GPT-2 1.5B could give better results, it is difficult to run the experiments to verify this.
>
> - Parsing - how did the authors deal with examples that do not syntactically parse in the Java-C# dataset?
>     - The tree sitter parser still gives a tree but with error nodes for such cases. We treated the error node like any other node in the syntax tree for our experiments.
>
> - AST Path Prediction - are all nodes in a single path predicted at the same time, independently of each other?
>     - Yes, all nodes are predicted at once independently. We used a separate linear layer to predict the node type at each level. We avoided using a more complicated module like RNN to minimize the computation overhead from the auxiliary task.
>
> - We edited the first paragraph in the introduction and added citations.
> - We increased the font sizes in Figure 1 in the revision.

---

### Review · Reviewer_1517 · 2022-11-19

**Summary Of Contributions:**

The main contribution of the paper is the additional attention layers in both encoder and decoder based on properties of data flow and AST structures on top of the standard self-attention mechanism that powers Transformer models.

**Audience:**

Yes

**Broader Impact Concerns:**

The paper has addressed the concerns on the ethical implications of the work sufficiently.

**Claims And Evidence:**

Yes

**Requested Changes:**

Addressing every single point in **Cons** described above

**Strengths And Weaknesses:**

## Pros

- Novel attention mechanism based on AST and dataflow.
- Extensive evaluation.

## Cons
- **overblown claims**: starting from *Code translation has applications in ...* all the way to *in speeding up the software development process*. Those points made are inaccurate, not only did authors provide no evidence in their evaluation that StructCoder is even remotely close to being helpful in real software development setting, but also this is contrary to the understanding of many (if not most) people in software engineering community that massive language models as they stand do not have the capability in writing real code in professional development setting.

- **issues with the design of StructCoder**: (1) the input of the models seems to contain redundancy, why is token sequence s1..sn and terminal nodes in the AST l1...lm are used at the same time? Considering that most of tokens are encoded in terminal nodes, and for the few that are not, you can simply augment the standard AST to include those, why not just use the terminal nodes of augmented AST? (2) In section 3.2.1, *we utilize node-type... to encode a node's semantics*, this is not true, given a method call like `a.foo()`, the AST node type does not at all encode the semantics of the method.

- **limitation of the approach**: the additional attention layers in encoder introduced in the paper are obviously not applicable to text-to-code generation tasks. Also, the paper already considers data flow properties to encode the code semantics, then it seems rather obvious to explore control dependency properties.

- **lack of details**: in equation (3) what does the symbol $\mathbb{1}$ mean? in section 3.3.1, when you predict the AST path as part of the learning task of decoder, how do you predict a sequence of node types. The paper says *we use a linear layer...* but this seems to give the prediction of only one node type rather than all in the path, so how is this done exactly, the common self-attention stuff or rnn like?

- **presentations issues**
  - in introduction *tradition code generation tools have ....using hand-crafted rules....* Citation please?
  - in 3.1 *Dij = 1if and only if value vi is obtained from vj* do you count direct dependency or kleene star as well?
  - in 3.1 *the translated code T solves exactly the same problem as...* this is so informal! At the very least, should use the term **semantic equivalence**
  - in 3.2.1 *use a defaut embedding for all DFG variables* what does it mean by *default*? fixed embedding that will not be learned? If so, why is that?

---

> ### Author Response · Authors · 2022-12-29
> **Author response**
>
> Thank you for your comments and questions. Here are our responses.
> - **overblown claims:** We edited the first para in the introduction and provided citations. We agree with the point that the current code LM works cannot write real code in a professional development setting. However, building these models is an important first step to this overall goal.
>
> - **issues with the design of StructCoder:** (1) We had planned to work on a similar idea like this in the future. For this paper, we chose to use separate code tokens and terminal nodes because a terminal node may be split into multiple tokens by the tokenizer, and we wanted StructCoder to behave like CodeT5 when no AST/DFG structures are input to the model in order to borrow the pretrained CodeT5 weights to minimize our pretraining. (2) Thanks for pointing it out. We corrected “node’s semantics” to “node’s syntax”.
>
> - **limitation of the approach:** We have not thought about control dependency properties of text for this work as we focus on code structures; we may explore it in the future. For StructCoder, we were using the structure related encoder components only when the input is code.
>
> - **lack of details:** $\mathbbm{1}$ denotes the indicator function. For predicting the sequence of node types, we predict the node type at each level using a separate linear layer on top of the same hidden embedding. We did not use a sequence model for this purpose.
>
> - We added citations to the first para in the introduction.
>
> - "in 3.1 Dij = 1if and only if value vi is obtained from vj do you count direct dependency or kleene star as well?"
>     - We just considered direct dependency here and added this clarification in section 3.1.
>
> - "in 3.1 the translated code T solves exactly the same problem as... this is so informal! At the very least, should use the term semantic equivalence"
>     - We rephrased the sentence using this suggestion.
>
> - "in 3.2.1 use a default embedding for all DFG variables what does it mean by default? fixed embedding that will not be learned? If so, why is that?"
>     - We used a special token embedding like that of <CLS> for the input embeddings of DFG variables. We changed the phrase to "and use a unique embedding for all DFG variables". This will be learned. We said "default" earlier to mean that it is the same for all DFG variables.

---

### Review · Reviewer_4t6u · 2022-11-30

**Summary Of Contributions:**

StructCoder is an encoder-decoder Transformer model for code. The main construct is to use code structure (ASTs and DFGs) in both the encoder and decoder modules. The model is trained with structure-aware encoder attention and denoising auto-encoding with AST path prediction and data flow prediction. It is compared against baseline methods on one code translation and two text-to-code generation datasets.

**Audience:**

Yes

**Claims And Evidence:**

Yes

**Requested Changes:**

* Section 3.1: The same variable can occur in multiple places. How is this handled? When you say $v_i$ is obtained from $v_j$, do you mean that this is a direct dependence or transitive? I believe you are using the same strategy as GraphCodeBERT here, but it is important to clarify this in the paper.
* Section 3.3.2: In a code snippet, you can have DFG edges going forward. As you are doing auto-regressive decoding, how do you handle this? Do you not need masking in Eq. (13)?
* You have explained what sequence lengths you have used for StructCoder in the appendix. Can you state what sequence lengths are used for the baselines for the respective tasks?
* StructCoder is finetuned for each of the tasks (Appendix C). Are baselines also finetuned?

**Strengths And Weaknesses:**

Strengths
----
* Though structure information has been used on the encoder side in the literature before, its use to train a decoder (in an encoder-decoder setup) is novel.
* The paper is easy to follow.
* The paper performs evaluation on existing datasets and baselines.

Weaknesses
-----
* The paper proposes novel modeling constructs to take advantage of structure of code. However, the experiments do not show substantial gains for the added complexity of training over CodeT5 even though StructCoder is initialized from CodeT5.
* On the CONCODE dataset, StructCoder reports EM of 22.35 which is slightly better than CodeT5 but is slightly below UniXCoder (22.60). UniXCoder is not included as a baseline.
* The encoder input consists of AST and DFG in addition to the code tokens. This increases the input size by ~2x and cost of self-attention by ~4x. Though the authors report (Section 4.3.4) that they haven't observed significant increase in inference time in their experiments, the sequence lengths considered by them are relatively small (Appendix C). The increase for longer sequence lengths > 2K can be substantial.
* Many generative models support code-to-text tasks which StructCoder does not support.
* Some of the technical and experimental details are not explained in the paper or the appendix. I request the authors to refer to the requested changes section for my questions.

---

> ### Author Response · Authors · 2022-12-27
> **Author response**
>
> Thank you for your comments and questions. Here are our responses.
>
> - Section 3.1: The same variable can occur in multiple places. How is this handled?
>
>     - This happens, for example, in Figure 1, where variable 's' occurs twice. Each occurrence is treated as a separate node in the DFG.
>
> - When you say $v_i$  is obtained from $v_j$, do you mean that this is a direct dependence or transitive? I believe you are using the same strategy as GraphCodeBERT here, but it is important to clarify this in the paper.
>
>     - Yes, we are using the same strategy as GraphCodeBERT.  It is a direct dependence between $v_i$ and $v_j$ and we added this clarification in the revision.
>
> - Section 3.3.2: In a code snippet, you can have DFG edges going forward. As you are doing auto-regressive decoding, how do you handle this? Do you not need masking in Eq. (13)?
>
>     - The decoder is masked like a regular Transformer decoder such that the hidden state $h_i$ is generated only from looking at target tokens before the $i^{th}$ position. We do auto-regressive decoding to get the hidden states. After obtaining all decoder hidden states, we use equation (12) to predict DFG edges. Hence, we do not need any masking in equation (13).
>
> - You have explained what sequence lengths you have used for StructCoder in the appendix. Can you state what sequence lengths are used for the baselines for the respective tasks?
>
>     - We added more info about the sequence lengths used by the most competitive baseline CodeT5, and a closely related baseline GraphCodeBERT, in Appendix C. The added text is printed in red in the revision.
>
> - StructCoder is finetuned for each of the tasks (Appendix C). Are baselines also fine-tuned?
>
>     - Yes, the baselines are fine-tuned as well. We borrow the numbers for most (except for fine-tuning CodeT5 on APPS which we did, and evaluating CodeT5 on CodeXGLUE translation task using their fine-tuned checkpoint) of the baselines from either CodeXGLUE leaderboard or previously published papers.

---

### Decision · Action_Editors · 2023-02-05

**Recommendation:** Reject

**Comment:**

While a number of concerns were addressed by the author response and post-review discussions amongst reviewers and AE (e.g., whether extra data was needed for the additional training or whether it came from the same pretraining set), there were still significant concerns remaining (see above).

**Audience:**

Yes, the paper is on a topic that would be of sufficient interest to TMLR's audience.

**Claims And Evidence:**

After discussion with reviewers, we unfortunately do not find key claims to be adequately supported:
* The improved performance is gained thanks to the reasons described in the paper
* This improved performance comes without a significant time penalty

For the first point, reviewers do not believe that the issues have been resolved around decoupling the structure modeling from the additional loss, and they would still like to see additional ablation studies. Further, reviewers are concerned about the use of different sequence lengths compared to the baselines, which makes it difficult to compare the results (and anyway show small improvements). Reviewers believe that more evidence and clear is needed to demonstrate the utility of the proposed design.

For the second point, reviewers asked for additional information about timing, but authors did not provide additional details regarding inference and training times.